# Psychological distress and burden of care among family caregivers of patients with mental illness in a neuropsychiatric outpatient clinic in Nigeria

**Ekerette Emmanuel Udoh**[1]*, **Deborah Eunice Omorere**[2], **Olarewaju Sunday**[3], **Olotu Sunday Osasu**[2], **Babatunde Abiodun Amoo**[1]

1 Society for Family Health, FCT, Abuja, Nigeria, 2 Federal Neuropsychiatric Hospital, Benin City, Nigeria, 3 Department of Community Medicine, Osun State University, Osogbo, Nigeria

* ekerette01@gmail.com

**Data Availability Statement:** All dataset files are available from the figshare database: https://doi.org/10.6084/m9.figshare.13348406.

## Abstract

### Background

The wellbeing of family caregivers of mentally ill persons has often been ignored, despite that family caregivers can be predisposed to psychiatric morbidities and burden in caring for their mentally ill family members. This study examined the levels of psychological distress and burden of care experienced by family caregivers who care for their mentally ill relatives in Edo State, Nigeria.

### Methods

This study assessed psychological distress using the General Health Questionnaire (GHQ-12). Burden of care was measured using the 22-item Zarit Burden Interview (ZBI) question-naire. Multiple linear regression was done to determine factors associated with burden of care and psychological distress, while factor analysis was used to determine the underlying forms of burden of care and psychological distress among participants.

### Results

Caregivers studied were relatives of patients diagnosed for depression (25.1%), substance use disorder (22.2%), schizophrenia (20.2%) and bipolar affective disorder (11.1%). Approximately 15% experienced no-to-mild burden, 51.3% mild-to-moderate burden and 34.0% high-or-severe burden. Nearly halve (49.0%) of participants experienced psychological distress. Severe rate of psychological distress was observed among subjects caring for patients with schizophrenia (60.7%), epilepsy (60.0%), substance use disorder (52.2%) and depression (49.0%). High burden of care was more preponderant among caregivers of relatives with mental retardation and epilepsy (50% each) and schizophrenia (39.3%). Having a higher educational qualification and being self-employed was a predictor of psychological distress. Gender of caregiver and the diagnosis schizophrenia among relatives of caregivers predisposed to burden of care. Three factors including social and emotional dysfunction,

**Funding:** The authors received no specific funding for this work.

**Competing interests:** The authors have declared that no competing interests exist.

psychological distress and cognitive dysfunction were identified as components of psychological health through factor analysis. On the burden scale, six factor components were identified as: personal strain, role strain, intolerance, patients' dependence, guilt and interference in personal life.

## Conclusion

There is a high prevalence of psychological morbidity and burden of care among family caregivers providing care for persons with mental illness.

## Introduction

Mental disorders are among the leading causes of morbidities and disabilities globally and estimates show that one-in-four families have at least one family member with a mental disorder [1]. At the end of 2001, 450 million people were suffering from a mental or behavioural disorder [2], however, in 2017 alone, about one billion people experienced one or more mental disorders, including substance use disorder [3]. Despite the high burden of mental disorders, there is an inadequate response with regards to treatment and provision of care for people with mental disorders. For instance, only 25% and 15% of people with mental disorders in low- and middle-income countries respectively receive treatment [4]. Available data in developing countries indicate a disproportionate rate of mentally ill patients who are receiving care from a medical professional or a professional caregiver. In addition, there is an extremely low availability of standard care homes or psychiatric facilities to cater for mentally ill people. Rather, people with mental illness regularly receive care and treatment in unprofessional settings, in homes by their family relatives, and evidence show that about 90% of people with mental illness get support from their families [5]. With these known challenges, there is a recognition of the importance and role family caregivers play in the long-term management of their mentally ill relatives. Conversely, owing to the prolonged course and chronic nature (compared to other forms of health conditions) associated with mental illnesses, family members are mostly compelled to take the responsibility of caregiving for their relatives with mental disorders. As a result, the debilitating impact caregivers experience during the process of caring for their mentally ill relatives have become a growing concern.

While caring by family members of a patient with mental disorder impact significantly on the patient's health, literature have documented the increasing concern about caregiver's wellbeing with respect to their psychological and physical health. Caregivers have been reported to experience psychological distress and burden during care of family relatives with mental disorders or disabilities. Researchers have therefore come to define caregiver burden of care as unwanted and negative experiences that caregivers experience as a result of taking care of their mentally ill relative [5]. Psychological distress on the other hand has been defined as a state of emotional suffering characterized by symptoms of depression (such as loss of interest, sadness and hopelessness) and anxiety (restlessness, feeling tense and somatic symptoms) that affects the individual's ability to cope with a particular set of circumstances [6]. Suffering psychological distress and burden during care, not only affect the quality of life and health of the caregiver, but will also affect their productivity as an individual, and their ability to provide quality care for the ill relative, therefore worsening the health of the mentally ill relative and decreasing the likelihood of their possible recovery or improved health.

Some studies have highlighted the extent of caregiver burden of care and psychological distress. An earlier study in Nigeria reported that about 45% of family caregivers who provided

care for schizophrenic relative experienced high burden of care [7]. In sub-Saharan countries, the burden of care among people living with mental illness has been reported to be high, ranging from 60–90% across different regions [5]. Meanwhile, in the developed regions, more than seven in ten (72%) caregivers of people living with mental illness experience significant burden. Similarly, evidence from low-and middle income countries indicated that a sizable proportion, 40%, of caregivers of mentally ill patients experience psychological distress [8, 9], Specifically, family caregivers have been reported of experiencing emotional, psychological, physical, social and financial difficulties because of caring for their mentally ill relative [7, 10]. In Nigeria, the scale and severity of the problem have been underrated, and care givers do not get any attention regarding their wellbeing and struggle while caring for the mentally ill. Furthermore, research evidence is also unavailable on the long-term impact of the experiences of burden and psychological distress in the caregivers' lives. This study seeks to determine the prevalence and determinants of psychological distress, and burden of care of family caregivers of relatives with mental disorders receiving outpatient treatment at a neuropsychiatric facility in Benin City, Edo State, Nigeria. We also examined the underlying forms of psychological distress and burden experienced by the caregivers.

## Methods

### Study setting

This was a descriptive hospital-based survey of relatives who are caregivers of patients suffering from mental illnesses. Information was collected from respondents attending the Federal Neuropsychiatric Hospital, Uselu in Benin City, Edo State in Nigeria. The centre is a tertiary health facility which receive referrals from primary and secondary health centres around the state and from other states in the South-South and South West geopolitical zones. The centre attends to an average of 450 existing and new patients load per week, and at least 150–200 of them attend with a family member who provides care for the patient. The hospital provides both in-patient and out-patient care. Eligible caregivers sampled in the study were identified as they accompanied their mentally ill relatives to the out-patient department (OPD) of the clinic for follow-up care. Eligible caregivers were male and female persons of age 18 to 65 years who have had close contact with the mentally ill relative and have lived with the patient for about 6 months. In addition, the patient of the recruited caregiver was expected to have been a patient of the hospital for at least six months. This study excluded the caregivers whose relative was still on admission in the hospital, and who did not report having any mental disorders themselves. Also, caregivers who did not live with the mentally ill relative for the specified duration of time, even if they assisted to bring their mentally ill relative to the hospital were not recruited.

The estimated number of relatives recruited for the study was determined using Cochran formula for estimation of single proportion, where $N = Z^2pq/d^2$ where Z = 1.96 (standard normal deviate); p = 43.8% (0.438) which is the prevalence of psychological distress in a study in Nigeria [9], q = 1-p; and d = Level of precision set at 0.05 (95% confidence interval). The minimum determined sample size was 378. Considering an anticipated non-response rate of about 10% in the study, a total of 415 participants were recruited in this study. Eligible caregivers were recruited consecutively by trained assistants on the survey days till the total sample size was reached. Data collection for this study lasted for three months, from January to March 2017.

Data was collected using a self-administered questionnaire. A translated version of the questionnaire from English to 'pidgin' English was administered in the situation where participants were unable to read the English version. However, in the situation where participants

were unable to read both the English and the 'pidgin' English version of the questionnaire, the trained research assistants conducted the interview with the participants by reading and ticking their responses of the participants using translated pidgin version.

## Measures

**Background characteristics of respondents.** Background information of the respondents including their socio-demographic characteristics and diagnosis of caregiver's relatives were elicited. They were age, marital status, employment status, monthly income, status of number of children, and the caregiver family relationship with the mentally ill patient. Diagnosis of the caregiver's relative was obtained from the clinical records of the patient and were matched to the participant/caregiver records.

**Burden of Care (BOC) schedule.** We assessed the burden of care of the participants using the Zarit Burden Interview (ZBI) questionnaire [11, 12]. The ZBI is a 22-item tool with 5-point likert scale (ranging from 0–4) that have responses as follows; never, rarely, sometimes, quite frequently and nearly always. Therefore, the overall summated scores of the 22 items range from 0–88. Zarit and colleagues [13] conceptualized burden as problems perceived by the caregiver with her or his health, psychological well-being, finances, social life, and the relationship between the caregiver and the ill family member.

**General Health Questionnaire-12.** The General Health Questionnaire version 12 (GHQ-12) was used to screen for probable psychiatric morbidity or psychological distress in the participants. This GHQ-12 is a self-administered screening instrument, sensitive to the presence of psychiatric disorders in individuals presenting in primary care settings and non-psychiatric clinical settings [14]. The GHQ-12 is a tool with 4-point scale (ranging from 0–3). Option of the GHQ-12 are as follows; not at all, no more than usual, rather more than usual, much more than usual. The overall summated scores range from 0–36.

## Data management and analysis

Data collected were coded and analyzed using IBM SPSS Version 20.0. Reliability analysis indicated an internal consistency of the Burden scale and the GHQ-12 with a Cronbach alpha score of 0.859 for the burden scale, and 0.506 for the GHQ-12 scale. A coefficient value between 0.50 and 0.70 is typically accepted as reliable [15]. The reliability measurement indicates whether the components of a tool is consistent with each other and can be reproduced using similar methodology and yield similar outcomes [15]. On the GHQ-12 scale respondents were awarded scores as follows; 0—Not at all, 1- no more than usual, 2- rather more than usual, 3- much more than usual. The mean of the population score (15.62±5.04) was used as cut-off to classify respondent as either experiencing psychological distress (scores greater than or equal to the mean) or not experiencing psychological distress (scores less than mean). On the burden of care schedule, computed scores award 0 for Never, 1- rarely, 2- sometimes, 3- quite frequently and 4- nearly always. Scores of two and above on each item suggested that participant experienced some burdens in the aspect of care giving. The total burden for each participant was the sum of scores on all items. Burden of care was classified as 'no to mild burden' (for total scores from 0–20), mild to moderate (scores 21–40), high or severe burden (scores > 40) [16]. Descriptive statistics was used to present all categorical and scale variables. Bivariate analysis was employed to test associations between background variables and the dependents variables of psychological distress and burden of care variables. A multiple linear regression using backward procedure was also conducted to model the predictive effect of the independent background variables, with the GHQ-12 and the Burden of care score. Correlational analysis was done to test relationship between the two dependent variables GHQ and

burden of care to indicate the construct validity of either tool, or to test the effect of one to the other. Level of statistical significance for all analysis was set at p<0.05.

**Factor analysis.** Factor analysis was performed to investigate the variable relationships of each scale of the GHQ12 variables and of the burden scale variables, to determine the underlying factors of the observations of the scale through their inter-correlations. To test the factorability of the data the Kaiser-Meyer-Olkin (KMO) test was used to determine the suitability of the data for factor analysis. The test measures sampling adequacy for each variable in the model and for the complete model. A KMO correlation above 0.50 is considered acceptable, while one above 0.90 is considered exceptional for analyzing factor analysis [17]. The Bartlett's Test of Sphericity was also employed to determine the appropriateness of the sample size for conducting the analysis. It tests the hypothesis that the correlation matrix under study is significantly different from the identity matrix, to indicate that the variables are related and therefore suitable for structure detection [17]. A significance level of 0.05 and below from the Bartlett's Test of Sphericity indicates that a factor analysis is useful with the data. To determine the factor structure in the factor analysis on the GHQ-12 and Burden of care variables, principal component analysis employing varimax rotation method was utilized. Factors were determined when eigenvalue greater than '1' was used. When the factor loadings for the components were lower than 0.3 the variable values were suppressed and therefore not displayed in the output. Any value greater than 0.30 are considered potentially meaningful factor loading results [15].

**Ethical considerations.** Ethical clearance and permission for the study was obtained from the research ethics review committee of the Federal Neuropsychiatric Hospital, Uselu in Benin City (PH/A.864/Vol. Vol, V11/201) before the survey was conducted. Research ethics principles were observed for an ethical conduct of the study. All participant voluntarily consented to take part in the study after being given adequate information about the study and signed the consent form. No participants' care was compromised by refusal to participate in the study.

## Results

Female caregivers of the mentally ill patient sampled in the study was approximately 57%, while male was 43.1%. Majority of participants in the study were in the age category 15–29 years (43.6%), next to respondents aged 30–44 years (37.1%), while participants from 45 years old and older accounted for 19.3%. More than halve (52.3%) of the participants were single while another majority were married (41.4%). About 48% of the caregivers had their own children. Majority of the relatives understudied were caregivers of patients diagnosed in the clinic with depression (25.1%), substance abuse disorder (22.2%), schizophrenia (20.2%) and bipolar affective disorder (11.1%). Mental retardation and epilepsy were 7.7% and 2.4%, respectively. Other grouped diagnoses were anxiety, obsessive compulsive neurosis, mania, hysteria etc (11.3%). Table 1 present details of the socio-demographics of the caregiver and clinical diagnosis of their relative.

The total maximum score in the population of study on the burden scale was 88, and mean score was 35.46.20±13.74. The total maximum GHQ-12 score recorded in the study population was 32 with a mean of 15.62±5.04. Nearly halve of the participants experienced psychological distress (49%). On the burden of caregiving, 14.7% experienced: no-to-mild burden, 51.3%; mild-to-moderate burden, and 34.0% experienced high burden. There was a positive correlation (r = 0.295, p = 0.000) between the participants' psychological distress with their burden of care.

Our study showed that there was a higher degree of burden of care among females than males in caring for their mentally ill relative. An apparently lower psychological distress

**Table 1. Background characteristics of the caregivers.**

| Characteristics | No. | % |
|---|---|---|
| **Sex** | | |
| male | 179 | 43.1 |
| female | 236 | 56.9 |
| **Age** | | |
| 15–29 years | 181 | 43.6 |
| 30–44 years | 154 | 37.1 |
| 45–59 years | 63 | 15.2 |
| Above 60 years | 17 | 4.1 |
| **Marital status** | | |
| Single | 217 | 52.3 |
| Married | 172 | 41.4 |
| Separated | 14 | 3.4 |
| Widowed/widower | 12 | 2.9 |
| **Religion** | | |
| Christianity | 362 | 87.2 |
| Islam | 48 | 11.6 |
| **Traditional** | 5 | 1.2 |
| Level of education | | |
| No formal education | 28 | 6.7 |
| Primary education | 66 | 15.9 |
| Secondary education | 112 | 27.0 |
| Higher education | 209 | 50.4 |
| **Employment status** | | |
| Unemployed | 120 | 28.9 |
| Self employed | 129 | 31.1 |
| Civil servant | 114 | 27.5 |
| Student | 32 | 7.7 |
| Others | 20 | 4.8 |
| **Monthly earning in Naira** | | |
| <18,000 | 18 | 12.9 |
| 18,000–30,000 | 37 | 26.6 |
| 31,000> | 84 | 60.4 |
| **Have children** | | |
| None | 216 | 52.0 |
| Yes | 199 | 48.0 |
| **Number of children** | | |
| 1–2 children | 53 | 26.6 |
| 3 children and more | 146 | 73.4 |
| **Relationship with patient** | | |
| Parent | 117 | 28.2 |
| Sibling | 104 | 25.1 |
| Uncle/aunt | 48 | 11.6 |
| Cousin | 44 | 10.6 |
| Spouse | 46 | 11.1 |
| Others | 56 | 13.5 |
| **Number living in household** | | |
| 3 and less | 63 | 16.2 |

(*Continued*)

**Table 1.** (Continued)

| Characteristics | No. | % |
|---|---|---|
| 4–6 | 144 | 37.1 |
| 7+ | 181 | 46.6 |
| **Diagnosis of caregiver's relative** | | |
| Schizophrenia | 84 | 20.2 |
| Bipolar affective disorder | 46 | 11.1 |
| Depression | 104 | 25.1 |
| Substance abuse disorder | 92 | 22.2 |
| Mental retardation | 32 | 7.7 |
| Epilepsy | 10 | 2.4 |
| Others | 47 | 11.3 |

(p>0.05) was observed for females than males (See Table 2). Mean burden of care was higher among older relatives aged 45–59 and above, compared with those aged 44 years and below. On the psychological distress scale, the reverse was the case. The mean psychological distress experience was higher among participants aged 44 years and below compared with those aged 45 years and older. There was a significant difference (P<0.005) in the mean burden of care with respect to marital status. A higher mean burden of care was observed among participants categorized as separated (44.28±10.76) and widowed (45.41±11.88), compared with the married (34.97±14.01) and participants who were single (34.73±13.46). With respect to the diagnosis of the patients of the caregivers, a significant difference in means of burden of care and of psychological distress was seen. The caregivers whose patient suffered epilepsy (38.60±11.68) and mental retardation (38.59±14.36) showed higher mean in experience of burden compared with the rest of the diagnoses such as depression (35.03±14.67) and bipolar affective disorder (32.84±10.51).

Fig 1 presents the prevalence of burden of care according to the different diagnosis of caregivers' relatives. Participants caring for relatives with epilepsy and mental retardation showed 50% each for high or severe burden of care. Similarly, with respect to psychological distress, patients with schizophrenia (16.30±4.59), epilepsy (17.10±1.91), and substance use disorder (16.33±4.02) presented higher levels of psychological distress in the study. Fig 2 shows the prevalence of psychological distress experience according to diagnosis of caregiver's relatives. Very high prevalence of psychological distress was seen among participants caring for relatives suffering schizophrenia (60.7%), epilepsy (60.0%) and substance use disorder and depression (52.2% and 49.0%, respectively).

Multivariate linear regression analysis using backward procedure showed that the main predictors of psychological distress were educational status, employment status and burden of care in the participants (Table 3). Similarly, the main predictors of burden of care among the participants were diagnosis of caregiver's relative, sex of the respondents, marital status, and psychological distress in the participants (Table 4).

In the Factor analysis the KMO and Barlett's test on both the GHQ-12 and the burden of care scale indicate that our data is suitable for the factor models, and the sample size appropriate for analysis (Table 5). The factor analysis (Table 6) presented three components of the psychological health scale which we can classify into (1) social and emotional dysfunction, (2) psychological distress and (3) cognitive dysfunction. The three components presented an overall variance of 49.45%. The items that were labelled as 'social and emotional dysfunction' included five items namely: able to concentrate on what you are doing, capable of making decisions, enjoy normal day to day activities, able to face up problems and feeling reasonably

**Table 2. Test of difference in mean for burden of care and psychological distress score.**

| | Burden of care score | Test statistic | p-value | Psychological distress score | Test statistic | p-value |
|---|---|---|---|---|---|---|
| | Mean±SD | | | Mean±SD | | |
| **Overall mean** | 35.46±13.74 | | | 15.62±5.04 | | |
| **Sex** | | | | | | |
| Male | 34.09 ± 11.95 | 3.154 | 0.076 | 15.95±4.84 | 1.378 | 0.241 |
| Female | 36.50 ±14.90 | | | 15.36 ±5.18 | | |
| **Age** | | | | | | |
| 15–29 years | 34.29 ±13.17 | 2.39 | 0.068 | 15.61±4.95 | 1.231 | 0.298 |
| 30–44 years | 35.09 ±12.81 | | | 16.07±5.02 | | |
| 45–59 years | 38.09± 15.84 | | | 14.87±5.12 | | |
| Above 60 years | 41.58± 17.62 | | | 14.35±5.72 | | |
| **Marital status** | | | | | | |
| Single | 34.73±13.46 | 4.398 | 0.005 | 15.62±4.98 | 0.185 | 0.907 |
| Married | 34.97 ±14.01 | | | 15.56±5.22 | | |
| Separated | 44.28 ±10.76 | | | 16.57±3.39 | | |
| Widowed/widower | 45.41 ±11.88 | | | 15.33±5.59 | | |
| **Religion** | | | | | | |
| Christianity | 34.95±13.66 | 2.092 | 0.125 | 15.51±5.08 | 1.170 | 0.311 |
| Islam | 38.67 ±14.40 | | | 16.10±4.34 | | |
| Traditional | 41.80 ±8.67 | | | 18.60±7.82 | | |
| **Level of education** | | | | | | |
| No formal education | 37.25±7.05 | 7.024 | 0.000 | 17.00±3.23 | 7.234 | 0.000 |
| Primary education | 40.78 ±13.79 | | | 16.89±4.31 | | |
| Secondary education | 37.02 ±12.90 | | | 16.60±4.80 | | |
| Higher education | 32.71 ±14.24 | | | 14.50±5.35 | | |
| **Employment status** | | | | | | |
| Unemployed | 34.81± 12.65 | 0.472 | 0.756 | 16.66±4.40 | 2.640 | 0.033 |
| Self employed | 35.29 ±13.97 | | | 14.99±5.83 | | |
| Civil servant | 36.81 ±13.77 | | | 15.78±4.55 | | |
| Student | 35.15 ±16.36 | | | 14.56±5.24 | | |
| Others | 33.30 ±14.63 | | | 14.15±4.61 | | |
| **Monthly earning in Naira** | | | | | | |
| <18,000 | 39.94 ±11.05 | 5.395 | 0.006 | 15.94± 4.37 | 0.817 | 0.444 |
| 18,001–30,000 | 39.72 ±14.48 | | | 15.89± 5.24 | | |
| 30,001> | 32.21 ±13.21 | | | 14.75± 5.38 | | |
| **Have Children** | | | | | | |
| None | 34.03 ±12.94 | 4.924 | 0.027 | 15.33± 5.03 | 1.473 | 0.224 |
| Yes | 37.02 ±14.44 | | | 15.93± 5.04 | | |
| **Number of children** | | | | | | |
| 1–2 children | 34.88 ± 12.94 | 1.430 | 0.210 | 16.81 ± 4.35 | 4.999 | 0.140 |
| 3 children and more | 37.79 ± 14.91 | | | 15.61 ± 5.25 | | |
| **Relationship with patient** | | | | | | |
| parent | 35.29± 12.39 | 1.580 | 0.164 | 16.29± 5.27 | 2.295 | 0.045 |
| sibling | 34.25± 11.52 | | | 14.87± 4.83 | | |
| uncle/aunt | 38.68 ±13.89 | | | 16.75 ±5.18 | | |
| Cousin | 36.43± 14.40 | | | 15.47 ±4.64 | | |
| spouse | 37.97 ±17.22 | | | 16.19±4.03 | | |
| others | 32.48 ±15.88 | | | 14.28±5.52 | | |

(*Continued*)

**Table 2.** (Continued)

| | Burden of care score | Test statistic | p-value | Psychological distress score | Test statistic | p-value |
|---|---|---|---|---|---|---|
| | Mean±SD | | | Mean±SD | | |
| **Number living in household** | | | | | | |
| 3 and less | 38.50± 14.29 | 2.519 | 0.082 | 15.30±4.77 | 4.500 | 0.012 |
| 4–6 | 34.04± 14.70 | | | 14.82±5.67 | | |
| 7+ | 34.66± 12.29 | | | 16.47±4.56 | | |
| **Diagnosis of caregiver's relative** | | | | | | |
| Schizophrenia | 37.96± 12.26 | 3.058 | 0.006 | 16.30±4.59 | 2.376 | 0.029 |
| Bipolar affective disorder | 32.84± 10.51 | | | 15.84±6.33 | | |
| Depression | 35.03± 14.67 | | | 15.36±5.58 | | |
| Substance abuse disorder | 36.82± 12.66 | | | 16.33±4.02 | | |
| Mental retardation | 38.59± 14.36 | | | 15.03±5.39 | | |
| Epilepsy | 38.60 ±11.68 | | | 17.10±1.91 | | |
| others | 29.06± 16.81 | | | 13.42±4.65 | | |

Note: SD = Standard deviation

happy). The component: 'psychological distress' was made up of four items as follows: couldn't overcome difficulties, feeling unhappy and depressed, losing confidence in yourself, and thinking of yourself). The 'cognitive dysfunction' component was made up of three items viz: (lost much sleep, felt you are playing a useful part in things and constantly under strain). On the Burden scale, six factor components were identified through the factor analysis as (1) personal strain (2) role strain (3) intolerance (4) patients' dependence (5) guilt and (6) interference in personal life. These 6 factor components explained a variance of 58.15% (Table 7).

## Discussion

In this study, depression, schizophrenia, and substance use disorder were the main mental disorders of patients of family caregivers, as about two-third of the caregiver's patients had one of the three conditions. Globally, these three conditions are among the top mental disorders [18].

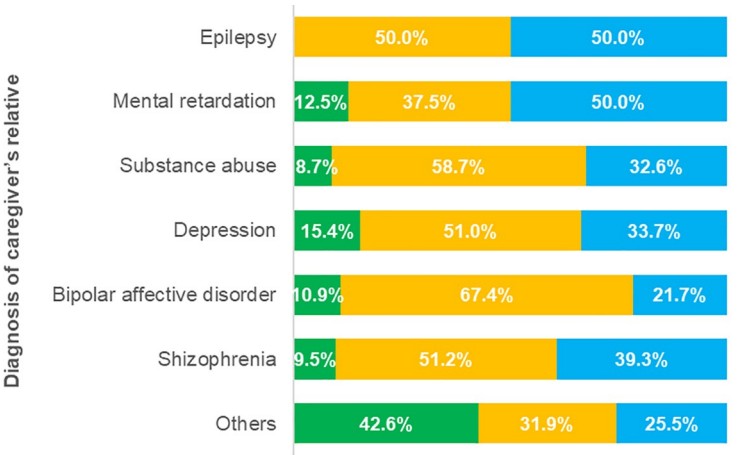

**Fig 1. Prevalence of caregiver burden of care according to caregiver's patient diagnosis.**

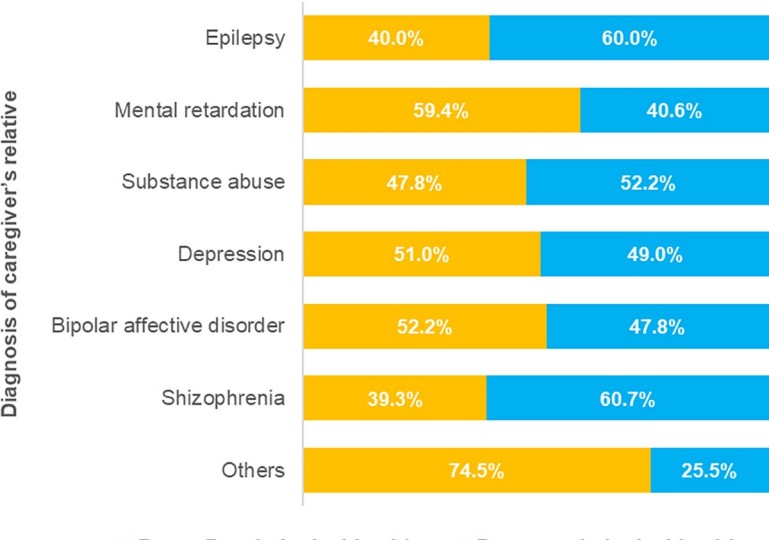

**Fig 2. Psychological distress experience by to caregiver's patient diagnosis.**

Lasebikan et al. [19] also found that these mental illness types are among the commonest in Nigeria. In the present study, prevalence of burden of care, 85.3%, consisting of 51.3% being mild-to-moderate and 34.0% being high/or severe burden, is comparable to a similar study done in Chile which reported 90.2% of burden among caregivers of mental disorder [20]. In other studies done in Southwest Ethiopia and in New Delhi, India, lower rates of burden among caregivers of people with mental illness, 72.9% and 75.1%, respectively, representing moderate to severe burden were reported [5, 21]. Mean burden of care in our study, was lower (35.46±13.74) compared with a similar study in Lagos Nigeria, with a mean of 41±18.6 [9]. The study utilised the same burden schedule with our study and was conducted among relatives of psychiatric patients but with a lesser sample size.

In the present study, experience of psychological morbidity among caregivers was rife, as nearly halve of the caregivers presented with psychological distress (49%). This rate is higher compared with the rate in a similar study in Malaysia were 31.5% of the caregivers have been reported of experiencing mild to great psychological distress in caring for patients with schizophrenia [22]. However, in Ethiopia, the overall prevalence of psychological distress among

**Table 3. Linear regression for the predictors of psychological distress during care of mentally ill.**

|  | Standard. Error | Standardized beta | t | Sig. |
|---|---|---|---|---|
| Religion: Christianity vs Traditional | 2.516 | -.268 | -1.763 | .080 |
| Religion: Islamic religion vs Traditional | 2.767 | -.243 | -1.671 | .097 |
| Educational status: Higher education vs No education | .868 | -.230 | -2.775 | .006 |
| Employment status: Self-employed vs others | .984 | -.238 | -2.907 | .004 |
| Employment status: Student vs others | 2.691 | -.129 | -1.696 | .092 |
| Diagnosis of caregiver's relative: Depression vs others | .993 | .136 | 1.700 | .092 |
| Burden of care | .031 | .293 | 3.672 | .000 |

Notes: t = t statistic, Sig. = significance level. Model information: R = 0.524, $R^2$ = 0.275, Model fit: F = 6.867, P-value = 0.000. The following background variables sex, age, marital status, religion, level of education, employment status, monthly earning, have children, relationship with patient, number of living children, diagnosis of caregiver's relative and the psychological distress variable were included in the first model in the backward procedure in the model.

**Table 4. Linear regression for the predictors of Burden during care of mentally ill.**

| | Standard. Error | Standardized beta | t | Sig. |
|---|---|---|---|---|
| Sex: Male | 2.391 | -.217 | -2.802 | .006 |
| Age | .094 | .136 | 1.753 | .082 |
| Marital status: Separated vs Widowed/widower | 4.199 | .174 | 2.368 | .019 |
| (Employment status) Self-employed vs others | 2.451 | .148 | 1.885 | .062 |
| Number in household | .155 | -.144 | -1.919 | .057 |
| Diagnosis of caregiver's relative: Schizophrenia | 2.295 | .288 | 3.603 | .000 |
| Psychological distress | .198 | .315 | 4.136 | .000 |

Notes: t = t statistic, Sig. = significance level. Model information: R = 0.575, $R^2$ = 0.331, Model fit: F = 8.976, P-value = 0.000. The following background variables sex, age, marital status, religion, level of education, employment status, monthly earning, have children, relationship with patient, number of living children, diagnosis of caregiver's relative and the psychological distress variable were included in the first model in the backward procedure in the model.

caregivers of patients with severe mental illnesses was reported to be higher, approximately 56.7% than in our study [23]. The levels of burden and psychological distress in our study were higher compared with rates found among caregivers of patients with non-psychiatric medical condition in a study in Nigeria, were 35% experienced psychological distress and 11% reported experiencing high burden of care amongst caregivers of patients with Type-2 diabetes mellitus [24]. This signifies the severity of the burden and psychological distress in caring for patients with psychiatric disorders.

Different diagnosis of caregivers' relatives, significantly characterised psychological distress experiences of the caregivers. Different psychiatric illnesses portend different levels of burden and psychological distress due to their varied distressing symptoms [25]. We found that burden of care and psychological distress were significantly higher among caregivers whose relatives had mental retardation, epilepsy, and schizophrenia. The high burden on caregivers from these illnesses can be attributed to the fact that these disorders are chronic disorders with life time morbidities [26, 27], with a tendency of inflicting greater burden to manage by caregivers. For instance, studies have shown that schizophrenia and epilepsy, are among the most difficult mental illnesses to manage [28, 29]. In addition, mental retardation has also been associated with enormous physical and emotional burden especially among parents of children suffering from it [30]. Our findings highlight the consequence of psychological effect of the burden of caring for chronic mental illnesses.

Our findings that burden of care was higher among female family caregivers compared with males, corroborates with other studies in Ethiopia [5] and India [31]. Females are naturally inclined to assume the role of caregivers for generally ill or mentally ill family members and relatives. In addition, female caregivers have more emotional, social, financial and relationship burden and therefore more prone to burden of care [5], unlike, males who tend to have more 'managerial style' attitude to distance themselves from the stressful situations [32].

**Table 5. KMO and Bartlett's Test on the General Health Questionnaire 12 and the burden of care.**

| Kaiser-Meyer-Olkin and Bartlett's Test | | General Health Questionnaire | Burden of Care |
|---|---|---|---|
| Kaiser-Meyer-Olkin Measure of Sampling Adequacy. | | 0.674 | 0.853 |
| Bartlett's Test of Sphericity | Approx. Chi-Square | 857.580 | 2593.208 |
| | Df | 66 | 231 |
| | Sig. | 0.000 | 0.000 |

Note. Df = Degree of freedom, Sig. = significance level

**Table 6. The extracted components and factor loadings based on a principal components analysis with Varimax rotation for General Health Questionnaire 12.**

**Rotated Component Matrix**

| | Component | | |
|---|---|---|---|
| | **Factor 1: Social and Emotional Dysfunction** | **Factor 2: Psychological distress** | **Factor 3: Cognitive** |
| Able to concentrate on what you are doing | .517 | | |
| lost much sleep | | | .448 |
| Felt you are playing a useful part in things | | | .523 |
| Capable of making decisions | .619 | | |
| Constantly under strain | | | .719 |
| Could not overcome difficulties | | .524 | |
| Enjoy normal day to day activities | .789 | | |
| Able to face up problems | .772 | | |
| Feeling unhappy and depressed | | .570 | |
| Losing confidence in yourself | | .819 | |
| Thinking of yourself | | .761 | |
| Feeling reasonably happy | .461 | | |
| % Variance | 21.12 | 17.77 | 10.56 |
| Eigenvalue | 2.53 | 2.13 | 1.26 |

Our study tends to show a higher degree of burden of care with increase in age. This was similar with a study in India [31]. More burden among the older people, might be because older people assume more responsibilities in care than younger people, for example in providing for the financial, social and medical needs of their ill relatives. These older people may also be the guardians or parents of the mentally ill persons, and so tend to receive the impact of the poor health of their relatives more [30]. In addition, older age is typically related to depression [33].

Being self-employed presented as a significant predictor of psychological distress. Caring for mentally ill family relative may become a burden on caregivers due to limited financial resources and failure to meet other responsibilities and commitments. This will most likely impact the quality of care which they are able to provide and consequently their own health [34, 35]. These findings are predictable, as socio-economic status based on employment, and education are generally determinants of health and mental health specifically [36]. Although this study did not explore whether the unemployed respondents lost their work as a result of the need to care, there are indications that this could be a likelihood.

We observed that the GHQ-12 tool and the Zarit Burden of care schedule adequately assessed the psychological health and burden of caregivers, respectively, as the Cronbach alpha values showed for the measures. The factor analysis result identified three components of psychological health. These items comprised of similar factors reproduced in the observed three-factor mode in other studies in Malaysia and Saudi Arabia [15, 37]. However, a different nomenclature was used in the study by El-Metwally et al. [37] for the identified three components of psychological health which were Social dysfunction, anxiety and loss of confidence.

On the burden of care tool, we identified six forms of burden to include personal strain, role strain, intolerance, patient's dependence, guilt and interference in personal life. While several studies have been found to identify between 2 to 5 factors on the Zarit Burden of care interview [38, 39], no study has yet found and presented 6 factors from the burden scale. The studies that identified two, or three factors, found: personal strain, role strain [40], and guilt (or dementia) [41]. Other combination of factors that have been identified are patients' dependency, self-criticism and embarrassment and anger [42]. One study that identified up to five component labelled the components as negative emotion, interpersonal relationship, time

**Table 7. The extracted components and factor loadings based on principal components analysis with Varimax rotation for the burden of care scale.**

**Rotated Component Matrix**

| | Components | | | | | |
|---|---|---|---|---|---|---|
| | Factor 1: Personal strain | Factor 2: Role strain | Factor 3: Intolerance | Factor 4: Patient's dependence | Factor 5: Guilt | Factor 6. Interference in personal life |
| Felt relative ask for more help than he/she needs. | | | | | | .717 |
| Felt that because of time spent with relative there is not enough time for self. | | | | | | .749 |
| Felt stressed between caring for self and trying to meet other responsibilities for family work? | .330 | | | | | .331 |
| Felt embarrassed over relative's behaviour. | .552 | | | | | |
| Felt angry when around relative. | .706 | | | | | |
| Felt Relative currently affects relationship with other family members or friends in a negative way. | .732 | | | | | |
| Afraid of what future holds relative. | .652 | | | | | |
| Felt relative is dependent on me. | .334 | | | .719 | | |
| Felt strained when around relative. | .394 | .313 | | .575 | | |
| Felt health has suffered because my involvement with relative. | .357 | .616 | | | | |
| Felt less privacy as would like because of relative. | | .797 | | | | |
| Felt social life has suffered because of caring for relative? | | .780 | | | | |
| Felt uncomfortable about having friends over because of relative? | | .607 | | | | .314 |
| Felt relative seems to expect to be cared for as the only one he/she could depend on. | | | | .662 | | .326 |
| Felt there's not enough money to take care of relative in addition to the rest of personal expenses. | | | .302 | .589 | | |
| Felt will be unable to take care of relative much longer. | | | .648 | | | |
| Felt loss of control over life since relative's illness. | .344 | | .666 | | | |
| Felt uncertain about what to do about relative. | | | .676 | | | |
| Wished to leave the care of relative to someone else. | | | .703 | | | |
| Felt should be doing more for relative. | | | | | .846 | |
| Felt could do a better job in caring for relative. | | | | | .859 | |
| Burdened about caring for relative. | | | | | .439 | |
| % Variance | 26.80 | 9.44 | 6.30 | 5.91 | 5.09 | 4.61 |
| Eigenvalue | 5.89 | 2.07 | 1.38 | 1.30 | 1.12 | 1.01 |

demand, patient's dependence, self-accusation and guilt [39]. These component terminologies have some similarities with the constructs identified in our study.

It is important to note some of the limitations of this study. While it was relevant to research the rates of psychological distress and burden of care in several psychiatric diagnosis, the heterogeneous psychiatric conditions in the study may bias the overall rates of psychological distress and burden of care among participants. Also, this study only provides a snapshot of the level of psychological distress and burden of care in caregivers of psychiatric patients, but no information is derivable as to how severe distress and burden are, relative to other kinds of medical disorders. We, therefore, suggest future studies to compare the levels of psychological distress and burden of care in both psychiatric and non-psychiatric patients. Other limitations are that the sample for this study was from one geographic location and a in single facility in Edo State. In addition, the location of the facility is in an urban and high-income area of the

state, therefore, generalization of the findings should be done with caution regarding caregivers in other regions such as rural or low-income areas. Also, the cross-sectional study design employed in this study limits causal inferences, as the study did not explore causal relationship between the caregiver characteristics and mental health status of their mentally ill family members or relatives. There is a possibility that there was no change in the observed family caregiver characteristics before onset of illness among their patients.

## Conclusion

In conclusion, experiences of psychological distress and burden of care among family caregivers of patients with mental illness is common and of a high prevalence. Caregivers caring for people with schizophrenia, mental retardation and epilepsy particularly experience high psychological morbidity and burden of caring for their ill relatives. Caregivers suffer social, emotional, psychological, and cognitive dysfunction as forms of psychological morbidity. Experiences of strain and interferences in their personal life from committing time and resources to the care of their ill relatives presents as some of the components of burden of caring for a mentally ill family relative. Health education to promote better and effective management of caregiving to limit exposure to psychological morbidity or mortality, and burden of care should be regularly taught to caregivers in some care support programmes. Advocacy for caregivers to be regularly screened for probable psychological morbidity and burden of care for early identification and intervention is needed.

## Acknowledgments

The authors are thankful to the respondents who were willing to take part in the study. We appreciate the management of the Federal Neuropsychiatric Hospital, Uselu, Benin City, for granting the permission to carry out this study. We appreciate the support received from the staff of the outpatient clinic during the data collection stages.

## Author Contributions

**Conceptualization:** Ekerette Emmanuel Udoh, Deborah Eunice Omorere, Olarewaju Sunday, Olotu Sunday Osasu.

**Data curation:** Ekerette Emmanuel Udoh, Deborah Eunice Omorere.

**Formal analysis:** Ekerette Emmanuel Udoh.

**Funding acquisition:** Deborah Eunice Omorere.

**Investigation:** Deborah Eunice Omorere, Olarewaju Sunday.

**Methodology:** Ekerette Emmanuel Udoh, Deborah Eunice Omorere, Olarewaju Sunday.

**Project administration:** Ekerette Emmanuel Udoh, Deborah Eunice Omorere.

**Resources:** Deborah Eunice Omorere.

**Supervision:** Ekerette Emmanuel Udoh, Olarewaju Sunday, Olotu Sunday Osasu.

**Validation:** Ekerette Emmanuel Udoh, Deborah Eunice Omorere, Olarewaju Sunday, Olotu Sunday Osasu.

**Visualization:** Ekerette Emmanuel Udoh.

**Writing – original draft:** Ekerette Emmanuel Udoh, Deborah Eunice Omorere, Olotu Sunday Osasu, Babatunde Abiodun Amoo.

**Writing – review & editing:** Ekerette Emmanuel Udoh, Deborah Eunice Omorere, Olarewaju Sunday, Olotu Sunday Osasu, Babatunde Abiodun Amoo.

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
