## [Decision Letter · Decision Letter 0]

8 Feb 2021

PONE-D-20-39002

Psychological Distress and Burden of Care among Family Caregivers of Patients with Mental Illness in a Neuropsychiatric Outpatient Clinic in Nigeria

PLOS ONE

Dear Dr. Udoh,

Thank you for submitting your manuscript to PLOS ONE. After careful consideration, we feel that it has merit but does not fully meet PLOS ONE’s publication criteria as it currently stands. Therefore, we invite you to submit a revised version of the manuscript that addresses the points raised during the review process.

We look forward to receiving your revised manuscript.

Kind regards,

Frédéric Denis, Ph.D.

Academic Editor

PLOS ONE

Journal Requirements:

Reviewers' comments:

Reviewer's Responses to Questions

**Comments to the Author**

1. Is the manuscript technically sound, and do the data support the conclusions?

Reviewer #1: Yes

Reviewer #2: Partly

2. Has the statistical analysis been performed appropriately and rigorously? 

Reviewer #1: Yes

Reviewer #2: Yes

3. Have the authors made all data underlying the findings in their manuscript fully available?

Reviewer #1: Yes

Reviewer #2: Yes

4. Is the manuscript presented in an intelligible fashion and written in standard English?

Reviewer #1: Yes

Reviewer #2: Yes

5. Review Comments to the Author

Reviewer #1: This study aims to determine the prevalence and determinants of psychological distress and caregiver burden for family caregivers of mentally ill relatives receiving outpatient treatment at a neuropsychiatric facility in Benin City, Edo State, Nigeria. With the reduction in the number of hospital beds and the increase in chronic psychiatric diseases, this is an important public health issue.

Data collection methods use international scales, including GHQ-12, (to screen for psychiatric morbidity or psychological distress of participants) and Zarit Burden's questionnaire, which seeks to assess with the caregiver their health, psychological well-being, finances, social life and the relationship between the caregiver and the ill family member.

The statistics are carried out according to high technical standards and are described in sufficient detail.For example, to determine the relevance of the data from the factor analysis: the Kaiser-Meyer-Olkin (KMO) and Bartlett test was also used to determine the adequacy of the sample size for the analysis.

The article presents relevant research findings which coroborate other international publications.While several of the studies identified between 2 and 5 factors in the Zarit Burden Caregiving Interview, this study is the only one that presented 6 factors on the Caregiving Burden Scale for family caregivers of patients suffering from depression, schizophrenia and substance use disorders ...

The conclusions are appropriately presented and supported by the data.The work is well articulated to the international literature. This study shows the shortcomings of the caregiver training system to deal with the problems inherent to caregiving.

Reviewer #2: The manuscript tackles one very relevant point, that is the psychological distress and burden of care of relatives of people suffering from mental illness, in a low/middle-income country where the rates of psychiatric patients accessing specialistic cares are disproportionately low compared to the needs of these individuals.

Despite this extremely valuable starting point, unfortunately the study design has some limitations and weakness that constraints me, although reluctantly, to propend for the rejection of the manuscript. Nonetheless, given the high social merits of the research, I'd rather indicate major revision, also to integrate my comments with those of the other reviewers and to allow Authors to respond my criticisms and, if they would be willing, to try to fix some issues I will raise.

I see three major limitations/weakness in the study.

1. The inclusion of heterogeneous psychiatric and even neurological (epilepsy) conditions. This fact biases the rates of psychological distress and burden of care among participants. Authors should have better done to focus only on one or a restricted number of conditions (possibly severe ones) or compare, if they had some rationale for doing it, distress/burden in one condition compared to another (for instance: one severe psychiatric diagnosis compared to another one or to a severe medical disorder)

2.50.4% of participants had a higher education degree. Is this rate consistent with education level in the general population? It will not in many western countries. I also noticed that more than 60% participants were in the higher monthly earning category. Why the rate of higher education participants was so high? Was this related to the kind of hospital where recruitment took place? Or to the geographical localization of the site (in a richest region of Nigeria, for instance)? Should this represent a challenge for generalization of findings to the Nigerian general population?

3. Only caregivers of psychiatric patients were included. While this allowed to capture the level of psychological distress and burden of care in this group of people, it adds nothing to the comprehension of how severe are distress and burden relatively to other kinds of medical disorders. Also, no comparison is given with other regions of the country, with other African or less-developed regions, or with well-developed countries. Therefore, the study only gives a photography of what happens in a limited geographical region, but it does not allow to locate the findings in a broader and more meaningful context, if not with indirect comparisons.

Additional comments.

1. Authors should indicate additional inclusion/exclusion criteria (if any) beyond the 18-65 age range. For instance, were relatives with mental disorders included? All type of relatives were included or only those who spent considerable daily or weekly time with the patient (just to figure out a possibility: once the patient was assisted at the appointment by a brother/sister not living with him/her, was the relative recruited?).

The 18-65 age range may be too stringent. A 70-year old father of 40-50-year old patient would have been excluded, but his psychological distress and burden of care may be expected to be extremely severe.

2. Authors should specify the dates when the recruitment was conducted

3. In the section "Background characteristics of respondents" what "status of number of children" is?

4. "When the factor loadings for the components were greater than 0.3" I believe Authors would intend lower than 0.3 in this point.

5. There are some grammar imprecisions, such as "Any value greater than 0.30 are". Authors should carefully revise English language

6. Factor analysis adds very few to the findings.

7. Discussion is too long. The Limitations paragraph accounts only for some potential biases and does not discuss the relevant limitations I have indicated above

6. PLOS authors have the option to publish the peer review history of their article (what does this mean?). If published, this will include your full peer review and any attached files.

Reviewer #1: **Yes: **Laurence FOND-HARMANT PhD-HDR

Reviewer #2: **Yes: **Felice Iasevoli

---

## [Author Response · Author response to Decision Letter 0]

13 Mar 2021

Reviewer #2: The manuscript tackles one very relevant point, that is the psychological distress and burden of care of relatives of people suffering from mental illness, in a low/middle-income country where the rates of psychiatric patients accessing specialistic cares are disproportionately low compared to the needs of these individuals.

Despite this extremely valuable starting point, unfortunately the study design has some limitations and weakness that constraints me, although reluctantly, to propend for the rejection of the manuscript. Nonetheless, given the high social merits of the research, I'd rather indicate major revision, also to integrate my comments with those of the other reviewers and to allow Authors to respond my criticisms and, if they would be willing, to try to fix some issues I will raise.

I see three major limitations/weakness in the study.

1. The inclusion of heterogeneous psychiatric and even neurological (epilepsy) conditions. This fact biases the rates of psychological distress and burden of care among participants. Authors should have better done to focus only on one or a restricted number of conditions (possibly severe ones) or compare, if they had some rationale for doing it, distress/burden in one condition compared to another (for instance: one severe psychiatric diagnosis compared to another one or to a severe medical disorder)

Author Response: The reviewer’s observation on the design of the study employing a heterogenous psychiatric condition is very valid. The authors have agreed with the reviewer’s suggestion to include it as a limitation of the study. Also, the points raised that study would have better focused on a restricted number of conditions or to compare distress and burden in different conditions, or another medical disorder has been given as recommendations for further study in the manuscript.

2.50.4% of participants had a higher education degree. Is this rate consistent with education level in the general population? It will not in many western countries. I also noticed that more than 60% participants were in the higher monthly earning category. Why the rate of higher education participants was so high? Was this related to the kind of hospital where recruitment took place? Or to the geographical localization of the site (in a richest region of Nigeria, for instance)? Should this represent a challenge for generalization of findings to the Nigerian general population?

Author Response: The rates of higher educational qualification found in our study is very plausible, given that the study was carried out in an urban area in Nigeria, and the facility is a tertiary hospital. We found other studies with similar high educational qualification. In a tertiary hospital-based study in an urban setting in Lagos up to 48% participants reported higher educational qualification .

Regarding ‘monthly earnings’, in our study we adopted the highest monthly earning category as the earnings above the national minimum income, which is N30,000 (which is about $65). Therefore, the high proportion (60%) in the highest monthly earning is plausible as many people earn more than the minimum wage of N30,000. We also see that a large proportion also earn less than the minimum wage, which is typical of the population.

Notwithstanding, the authors have emphasized caution on the generalization of the findings to the general Nigeria population.

3. Only caregivers of psychiatric patients were included. While this allowed to capture the level of psychological distress and burden of care in this group of people, it adds nothing to the comprehension of how severe are distress and burden relatively to other kinds of medical disorders. 

Author Response: The observation by the reviewer is very valid. We have cited a literature presenting rate of caregiver’s burden and psychological distress in a non-psychiatric medical condition in Nigeria for reference and comparison for their rates with findings of our study.

Also, no comparison is given with other regions of the country, with other African or less-developed regions, or with well-developed countries. Therefore, the study only gives a photography of what happens in a limited geographical region, but it does not allow to locate the findings in a broader and more meaningful context, if not with indirect comparisons.

Author Response: In the discussion section of our manuscript some evidence from studies in other developed and developing regions as reference for comparison with our findings have been provided.

Additional comments.

1. Authors should indicate additional inclusion/exclusion criteria (if any) beyond the 18-65 age range. For instance, were relatives with mental disorders included? All type of relatives were included or only those who spent considerable daily or weekly time with the patient (just to figure out a possibility: once the patient was assisted at the appointment by a brother/sister not living with him/her, was the relative recruited?).

Author Response: Two additional exclusion criteria that we employed have been included in the manuscript.

The 18-65 age range may be too stringent. A 70-year old father of 40-50-year old patient would have been excluded, but his psychological distress and burden of care may be expected to be extremely severe.

Author Response: This is a valid observation. We are, however, not able to make any revision in this regard.

2. Authors should specify the dates when the recruitment was conducted

Author Response: The dates of data collection have been added in the method section.

3. In the section "Background characteristics of respondents" what "status of number of children" is?

Author Response: The variable that shows the ‘status of number of children’ has been added in the descriptive and bivariate association results. In our initial analysis this variable was dropped, because we rather considered use of the variable ‘’Having children.’’ 

4. "When the factor loadings for the components were greater than 0.3" I believe Authors would intend lower than 0.3 in this point.

Author Response: This has been corrected to now read as ‘’lower than 0.3.’’

5. There are some grammar imprecisions, such as "Any value greater than 0.30 are". Authors should carefully revise English language

Author Response: The article has been carefully revised and some imprecise grammars have been corrected.

indicated

6. Factor analysis adds very few to the findings.

Author Response: The components from the factor analysis were numerous, thus, the authors limited their discussion to keep the article less lengthy. 

7. Discussion is too long. The Limitations paragraph accounts only for some potential biases and does not discuss the relevant limitations I have above

Author Response: Some less relevant points in the discussion section have been removed. We have also accepted all suggestions to include the study limitations raised by the reviewer.

---

## [Decision Letter · Decision Letter 1]

5 Apr 2021

Psychological Distress and Burden of Care among Family Caregivers of Patients with Mental Illness in a Neuropsychiatric Outpatient Clinic in Nigeria

PONE-D-20-39002R1

Dear Dr. Udoh,

We’re pleased to inform you that your manuscript has been judged scientifically suitable for publication and will be formally accepted for publication once it meets all outstanding technical requirements.

Kind regards,

Frédéric Denis, Ph.D.

Academic Editor

PLOS ONE

Additional Editor Comments (optional):

Reviewers' comments:

Reviewer's Responses to Questions

**Comments to the Author**

1. If the authors have adequately addressed your comments raised in a previous round of review and you feel that this manuscript is now acceptable for publication, you may indicate that here to bypass the “Comments to the Author” section, enter your conflict of interest statement in the “Confidential to Editor” section, and submit your "Accept" recommendation.

Reviewer #2: All comments have been addressed

2. Is the manuscript technically sound, and do the data support the conclusions?

Reviewer #2: Yes

3. Has the statistical analysis been performed appropriately and rigorously? 

Reviewer #2: Yes

4. Have the authors made all data underlying the findings in their manuscript fully available?

Reviewer #2: Yes

5. Is the manuscript presented in an intelligible fashion and written in standard English?

Reviewer #2: Yes

6. Review Comments to the Author

Reviewer #2: (No Response)

7. PLOS authors have the option to publish the peer review history of their article (what does this mean?). If published, this will include your full peer review and any attached files.

Reviewer #2: **Yes: **Felice Iasevoli

---

## [Editor Report · Acceptance letter]

23 Apr 2021

PONE-D-20-39002R1 

Psychological Distress and Burden of Care among Family Caregivers of Patients with Mental Illness in a Neuropsychiatric Outpatient Clinic in Nigeria 

Dear Dr. Udoh:

I'm pleased to inform you that your manuscript has been deemed suitable for publication in PLOS ONE. Congratulations! Your manuscript is now with our production department. 

Kind regards, 

on behalf of

Dr. Frédéric Denis 

Academic Editor

PLOS ONE